# The Functional Profile and Antioxidant Capacity of Tomato Fruits Are Modulated by the Interaction between Microbial Biostimulants, Soil Properties, and Soil Nitrogen Status

**DOI:** 10.3390/antiox12020520

**Published:** 2023-02-19

**Authors:** Paola Ganugi, Andrea Fiorini, Vincenzo Tabaglio, Federico Capra, Gokhan Zengin, Paolo Bonini, Tito Caffi, Edoardo Puglisi, Marco Trevisan, Luigi Lucini

**Affiliations:** 1Department for Sustainable Food Process, Università Cattolica del Sacro Cuore, Via Emilia Parmense 84, 29122 Piacenza, Italy; 2Department of Sustainable Crop Production, Università Cattolica del Sacro Cuore, Via Emilia Parmense 84, 29122 Piacenza, Italy; 3Department of Biology, Faculty of Science, Selcuk University, Konya Campus, 8300 Konya, Turkey; 4oloBion-OMICS LIFE LAB, 08028 Barcelona, Spain

**Keywords:** phenolic compounds, carotenoids, microbial biostimulants, antioxidant activity, metabolomics

## Abstract

The application of microbial biostimulants to plants has revealed positive effects related to nutrients uptake, stress tolerance, root development and phenological growth. However, little information is available exploiting the potential synergistic biostimulant action of microbes on the functional quality of the yields. The current research elucidated the effect of single or coupled action of biostimulants, associated with either optimal or reduced nitrogen application, on the functional quality of tomato fruits. Chemical assays and untargeted metabolomics were applied to investigate *Rhizoglomus irregulare* and *Funneliformis mosseae* administration (both being arbuscular mycorrhiza, AMF), under optimal or low N input conditions, alone or coupled to *Trichoderma atroviride* application. The coupling of AMF and *Trichoderma* fungal inoculations resulted in a synergistic biostimulant effect on tomato fruits under sub-optimal fertility, revealing improved concentrations of carotenoid compounds—B-carotene (0.647 ± 0.243 mg/100 g), Z-carotene (0.021 ± 0.021 mg/100 g), 13-z-lycopene (0.145 ± 0.052 mg/100 g) and all-trans-lycopene (12.586 ± 1.511 mg/100 g), and increased values for total phenolic content (12.9 ± 2.9 mgGAE/g), total antioxidant activity (phosphomolybdenum, 0.9 ± 0.2 mmolTE/g), radical scavenging activity (DPPH, 3.4 ± 3.7 mgTE/g), reducing power (FRAP, 23.6 ± 6.3 mgTE/g and CUPRAC, 37.4 ± 7.6 mg TE/g), and enzyme inhibitory activity (AChE, 2.4 ± 0.1 mg GALAE/g), when compared to control. However, evidence of carotenoid and bioactive compounds were exclusively observed under the sub-optimal fertility and no significant differences could be observed between the biostimulant treatment and control under optimal fertility.

## 1. Introduction

The use of plant biostimulants in agriculture has increased significantly over the last 10 years, mainly due to the successful advances in research that showed beneficial effects on plants, especially in terms of nutrient use efficiency, abiotic stress tolerance, quality traits and the availability of limited nutrients in the soil and rhizosphere [1,2,3].

More recently, an increasing number of experimental studies have addressed research evaluating the combined use of plant biostimulants with the aim of ensuring global food security and environmental sustainability without increasing the rate of nutrient use [4,5]. Antagonistic, additive, or rather synergistic interactions among different plant biostimulants categories have been described, depending on whether the combined effect was respectively less, equal, or greater than the effect obtained by each biostimulant individually [5]. In most cases, the combined application of microbial and non-microbial biostimulants has reflected a synergistic action on plants, revealing an increase in nutrient uptake, stress tolerance, root development and phonological growth [6,7,8,9]. However, only limited scientific literature is available regarding the effect of combined applications of biostimulants on fruit quality.

Tomato plant (*Solanum lycopersicum* L.) represents one of the most grown and consumed crops worldwide, mainly due to key role in the Mediterranean human diet as fresh or processed product. Given the commercial importance of this crop, research is addressing efforts targeted to improve the biochemical composition of the fruit, including the content of potentially health-beneficial components such as antioxidants (vitamin C, lycopene etc.). For this reason, various metabolomics approaches have been extensively adopted to gain a better understanding of the biochemical determinants concerning of fruit growth and quality, both under and presence and the absence of abiotic and biotic stress factors [10].

In this context, biostimulant products, which have been abundantly investigated for their multiple benefits for plants—including nutrient uptake and use efficiency stimulation, and abiotic/biotic stress tolerance increment, have been recently evaluated as a sustainable approach to improve food nutritional/functional values [11,12]. Concerning tomato, the latest advances in metabolomics have allowed for the elucidation of the physiological processes involved in fruit response to biostimulant application, which seem to be linked to higher levels of antioxidants, mineral nutrients (N, P, Ca, Na, Fe, Mn and Zn), total vitamin C and phenolics [13,14]. In particular, metabolic changes related to secondary metabolism have been previously observed in a well-establish symbiosis between tomato and two common mycorrhiza, *Rhizophagus irregularis* (*R. irregularis*) and *Funneliformis mosseae* (*F. mosseae*) [15,16]. Likewise, *R. irregularis* and *F. mosseae*-treated tomatoes have revealed higher content of total polyphenols, carotenoids, vitamins, and flavor compounds (sugars, titratable acids, volatile compounds) [17,18]. Nevertheless, the combined action of two or more biostimulants on tomato fruit quality remains few explored.

Given this premise, this paper gets insight into the effect of single or coupled action of biostimulants, associated with a high or low nitrogen (N) application, on the functional quality of tomato fruits. Specifically, the work makes use of chemical assays and untargeted metabolomics to describe and compare the impact of no treatment and low N input, with *Rhizoglomus irregulare* and *Funneliformis mosseae* administration, in conventional or low N input condition, either in single or coupled to *Trichoderma atroviride* application.

## 2. Materials and Methods

### 2.1. Plant Growth Conditions and Experimental Design

Two field experiments were conducted between May and August 2020 at “Pizzacchera S.n.c.” and “Felletti Luca” farms, respectively located near Parma (44°50′55.7″ N 10°15′34.4″ E) and Ferrara (44°49′49.8″ N 12°07′07.6″ E), Emilia-Romagna Region, Northern Italy. The two sites were selected as optimal (Pizzacchera) and sub-optimal (Felletti) soil fertility conditions. Initial soil properties in the 0–30 cm soil layer at optimal fertility were: % sand 18.2, % silt 48.5, % clay 33.3, organic matter 40.13 g kg^−1^, pH(H_2_O) 7.98, pH(CaCl_2_) 7.35, % CaCO_3_ 9.7, electrical conductivity (μS cm^−1^) 295, organic carbon (g kg^−1^) 23.28 and total soil Kjeldahl nitrogen (g kg^−1^) 2.55. Same properties in the same soil layer at sub-optimal fertility were: % sand 93, % silt 6, % clay 1, organic matter 8.43 g kg^−1^, pH(H_2_O) 7.54, pH(CaCl_2_) 7.09, % CaCO_3_ 5.8, electrical conductivity (μS cm^−1^) 346, organic carbon (g kg^−1^) 4.89 and total soil Kjeldahl nitrogen (g kg^−1^) 0.60.

In both farms, the experiment was performed on non-randomized plots with four pseudo-replicates, arranged to test biostimulant-based treatments on tomato fruits quality under conventional and low N input conditions. The single plot size was 144 m^2^ (30 m × 1.6 m). In details, five different treatments were compared: (1) conventional N input without microbial treatment (Control); (2) low N input (LowN); (3) mycorrhizal treatment (*R. irregulare* BEG72 and *F. mosseae* BEG234, 700 sp g^−1^ each species) under conventional N input (AMF); (4) mycorrhizal treatment under low N input (AMF + LowN); (5) mycorrhizal treatment coupled with *T. atroviride* application under low N input (AMF + *T.atroviride* + LowN). The microbial biostimulants were formulated commercial products supplied by Agrotecnogias Naturales (Tarragona, Spain), inoculated at transplanting according to label recommendations.

At harvest, tomato yields were determined by weighting tomato fruits from four randomly selected areas (25 m^2^ each) from each plot. In details, commercial (red ripe fruits), immature (green underripe barriers), overripe (homogenously rotten barriers) fruits, as well as fruits with apical rot fruits, were separated after removal of fruits from plant stems, and weighted separately. Then, yields were expressed as t/ha. The resulting values were divided by the total yield (t/ha) to respectively obtain the percentages of commercial, immature, rotten and apical rot yield. Finally, the mature yield of five plants per condition was collected and immediately frozen at −20 °C. Successively, the fruits were ground with liquid nitrogen using pestle and mortal for the following chemical analysis.

### 2.2. Carotenoids Determination

Carotenoids were determined by high performance liquid chromatography with diode array detection–mass spectrometry, using a 1290 Infinity II chromatograph (Agilent Technologies, Santa Clara, CA, USA) as previously reported [19]. Briefly, a binary elution using (A) methanol/acetonitrile/water (84:14:4, *v*/*v*/*v*) and (B) dichloromethane, with a 45 min gradient run (0 to 60% dichloromethane in 40 min) at 25 °C. Samples were extracted in ethanol:*n*-hexane (60:40, *v*/*v*) [20] and the chromatographic separation was achieved using a YMC carotenoids column (250 × 4.6 mm i.d., 5 µm particle size) and a flow rate of 1 mL/min. Detection was set at the wavelengths 450, 348 and 286 nm and quantification done against pure reference standards.

### 2.3. Preparation of Extracts for Total Phenolic, Flavonoids and Biological Activity Assays

The extracts were prepared using homogenizer-assisted extractions, 1 g of each sample was extracted in 20 mL of 80% methanol (*v*/*v*), using an Ultra-turrax (Ika, T25, Staufen, Germany) for 3 min. Then, the extracts were filtered, and the solvents were removed using rotary evaporator under vacuum at 40 °C. The extracts were stored at 4 °C until analysis.

### 2.4. Antioxidant and enzymatic assays

#### 2.4.1. Determination of Total Phenolics Content

Folin-Ciocalteu and AlCl_3_ assays, respectively, were utilized to determine the total phenolic and flavonoid contents [21]. Results were expressed respectively as gallic acid equivalents (mg GAEs/g extract) and rutin equivalents (mg REs/g extract). All experimental details are given in Appendix A.

#### 2.4.2. Determination of Antioxidant and Enzyme Inhibitory Effects

The antioxidant and enzyme inhibitory activity of the extracts was determined according to previously described methods [22]. DPPH and ABTS radical scavenging activity, cupric ion reducing antioxidant capacity (CUPRAC), and ferric ion reducing antioxidant power (FRAP) were expressed as mg Trolox equivalents (TE)/g extract. The metal chelating ability (MCA) was reported as mg EDTA equivalents (EDTAE)/g extract, whereas the total antioxidant activity (phosphomolybdenum assay, PBD) was expressed as mmol TE/g extract. AChE and BChE inhibitory activities were given as mg galanthamine equivalents (GALAE)/g extract; tyrosinase inhibitory activity was expressed as mg kojic acid equivalents (KAE)/g extract, and amylase inhibitory activities were presented as mmol acarbose equivalents (ACAE)/g extract. All experimental details are given in Appendix A.

### 2.5. UHPLC-ESI/QTOF-MS Untargeted Profiling of Tomato Fruits Polyphenols

Starting from the grounded samples, tomato fruits were extracted as previously described by [23]. Briefly, six replicates per thesis (2 gr each) were extracted in 20 mL of 80% methanol (*v*/*v*) acidified with 0.1% formic acid (*v*/*v*), using an Ultra-turrax (Ika, T25, Staufen, Germany). Later, the extracts were centrifuged (12,000× *g*) and 1 mL of the resulting supernatants was transferred into vials for the analysis. Two blank (BLK) samples, consisting of the methanolic solution used for the extraction, and seven quality control (QC) samples, prepared by pooling 20 µL from all supernatants, were also injected to respectively remove background interferences and assess the robustness of the analytical process.

Metabolite screening of tomato fruits was performed with an untargeted metabolomics approach, throughout a hybrid quadrupole-time-of-flight mass spectrometer coupled to an ultra-high performance liquid chromatographic system (UHPLC/QTOF). Specifically, a 1290 liquid chromatograph system, equipped with a binary pump and a Dual Electrospray JetStream ionization system, coupled to a G6550 mass spectrometer detector (Agilent technologies, Santa Clara, CA, USA) was used. 

Chromatographic separation of tomato compounds was carried out in reverse phase on an Agilent Poroshell 120 PFP column (100 mm × 2.1 i.d., 1.9 µm particle size), while the mobile phase consisted of ultrapure water (A) and acetonitrile (B), both acidified with formic acid (0.1%, *v*/*v*). A gradient elution mode was applied starting from 6% to 94% B in 32 min. The volume of injection was 6 μL for each replicate and the flow rate was 200 μL/min, according to [24].

The QTOF mass analyzer was set in positive move (ESI+) with the following ESI conditions: nitrogen as both sheath gas (12 L/min, 315 °C) and drying gas (14 L/min and 250 °C), 45 psi nebulizer pressure, 350 V nozzle voltage, 4.0 kV capillary voltage. The HR mass spectrometer worked in full scan mode (range 100–1200 *m*/*z*) at a range of 1 spectra/s with a resolution power of 30,000 full width at half maximum (FWHM) at *m*/*z* 200. Chromatogram processing was performed with the MassHunter Qualitative Analysis software (version B.06.00, Agilent Technologies, Santa Clara, CA, USA).

Afterwards, The Agilent Profinder B.07 software (Agilent Technologies, Santa Clara, CA, USA) was used to process the raw data mass features, according to a targeted ‘find-by-formula’ algorithm and basing on the Phenol-Explorer 3.6 database (http://phenol-explorer.eu, accessed on 28 October 2022). Following mass and retention time alignment, compound identification was based on both monoisotopic accurate mass and isotope pattern (accurate spacing and isotope ratio), adopting a mass tolerance of 5-ppm. Features which were not present in 100% of replications within at least one treatment were discarded.

According to the Phenol-Explorer subclass information, phenols were classified in the phenolic subclasses, whose cumulative intensities were calculated and converted in mg L-1 equivalent using standard solutions, as previously described by [25].

### 2.6. Statistical Analyses

One and Two-way analysis of variance (ANOVA) were carried out in Rstudio software in order to determine any statistically differences among tomato fruits when comparing different treatments, within and between the two farms. The elaboration was performed both for qualitative, productive, and chemical parameters. In the presence of statistically difference, Duncan test was conducted for multiple comparisons between pairs of treatments. 

The statistical analyses on the metabolomics dataset were performed using Mass Profiler Professional B.12.06 (Agilent technologies) software. Compound abundance was Log2 transformed and normalized at 75th percentile and baselined against the median. Firstly, unsupervised hierarchical cluster analysis (Squared Euclidean distance, Ward’s linkage rule) was carried out using the fold-change based heat map, to highlight the relatedness among treatments and the two farms.

Successively, for both farms, one-way ANOVA and Duncan test were carried out on semi-quantitative polyphenol results to determine whether there were any statistically significant differences between all and pairs of treatments.

## 3. Results

### 3.1. Yield Parameters

Results on yield parameters of tomato at sub-optimal fertility (Table 1) did not show any significant difference among treatments, except for the apical rot which proved to be considerably higher in the control samples (2.1 t/ha).

On the contrary, at optimal fertility, the reduction of nitrogen doses for LowN, AMF + LowN and AMF +*T. atroviride* + LowN led to a remarkable decrease in marketable production (Table 2). Additionally, microbial treatment did not reveal an increase of the total yield, having obtained the highest value for the control condition (104.9 t/ha).

### 3.2. Carotenoids Determination

Carotenoid content in tomato fruits was significantly different in the two farms—except for Z-carotene—highlighting higher concentrations of each carotenoid compound in optimal fertility samples (Appendix A). Moreover, for both cultivation sites, all-trans lycopene content was the highest, reaching the maximum average concentration of 12.586 ± 1.511 and 16.781 ± 1.797 mg/100 g extract respectively in AMF + *T. atroviride +* LowN-treated sub-optimal fertility samples and LowN-treated optimal fertility samples (Table 3 and Table 4). Interestingly, biostimulant application did not enhance carotenoid content in optimal fertility tomatoes, having found that, except for Z-carotene, LowN and Control treatments determined the greatest amounts of carotenoid compounds (Table 4). Contrary, in sub-optimal fertility samples, AMF + *T. atroviride* + LowN application showed the upmost concentrations of B-carotene, Z-carotene, All-E-y-carotene, 13-z-lycopene, while Cis-lycopene was mainly increased by AMF + LowN treatment (Table 3).

### 3.3. Total Bioactive Compounds Determination

The total content of phenolics and flavonoids in tomato fruits is provided in Table 5 and Table 6. Focusing on sub-optimal + optimal fertility samples, total phenolic content values ranged from 10.2 mg GAE/g extract to 15 mg GAE/g extract and the greatest amount (13.1 ± 2.2 mg GAE/g extract) was observed with Low N application, followed by AMF + *T. atroviride* + LowN treatment (12.7 ± 2.3 mg GAE/g extract) (Appendix A). Concerning total flavonoid content, two-way ANOVA revealed statically differences among treatments, ascribing to Control and AMF + LowN the highest concentrations, respectively of 0.9 ± 0.3 and 0.9 ± 0.2 mg GAE/g extract. Optimal fertility samples showed the topmost average value for both phenolics and flavonoids (Table 6).

### 3.4. Total Antioxidant Activity

The total antioxidant activity calculated via phosphomolybdenum [26] is provided in Table 5 and Table 6. Concerning this assay, total antioxidant abilities of optimal fertility tomato samples were significantly higher (0.9 ± 0.1 mmol TE/g extract) than those of sub-optimal fertility (0.8 ± 0.1 mmol TE/g extract) (Appendix A). However, keeping together the two farms, the study did not reveal any consistent difference among treatments. Curiously, looking at one-way ANOVA results, sub-optimal fertility samples enlightened a significant increment of total antioxidant activity following AMF + *T. atroviride* + Low N treatment (Table 5), while no treatment effect was pointed out for optimal fertility (Table 6). 

### 3.5. Radical Scavenging Activity

The free radical scavenging activity of tomato fruits was determined using DPPH and ABTS arrays and the results were presented in Appendix A, Table 5 and Table 6. Both assays are based on the quenching of these radicals through the transfer of either an electron or a hydrogen atom by antioxidant compounds. Concerning two-way ANOVA on DPPH assay data, AMF + *T. atroviride* + LowN treatment exhibited the highest activity (2.5 ± 2.9 mg TE/g extract) and no difference was observed between sub-optimal and optimal fertility farms (Appendix A). Contrary, two-way ANOVA on ABTS assay data demonstrated a remarkable higher radical scavenging activity for optimal fertility (25.7 ± 3.4 mg TE/g extract) and for LowN application (26.4 ± 4.2 mg TE/g extract), which was significantly different from AMF treatment. 

### 3.6. Reducing Power

The reductive ability reflects to the electron-donation ability of antioxidant compounds. The reductive ability of tomato extracts was measured with FRAP and CUPRAC assays, respectively aimed at quantifying the potential for reducing ferric to ferrous and cupric to cuprous ions [27]. Two-way ANOVA results showed a similar tendency for both assays, highlighting a superior reducing power for AMF + *T. atroviride* + LowN (21.6 ± 4.9 mg TE/g extract for FRAP and 35.1 ± 6.2 mg TE/g extract for CUPRAC) (Table 4). Regarding CUPRAC, this last treatment significantly differed from AMF + LowN, which revealed the lowest value (30.5 ± 2.4 mg TE/g extract). However, no significance was found for the farm factor, revealing a lack of effect linked due to the cultivation area.

### 3.7. Metal Chelating Activity on Ferrous Ions

Ferrous chelating activity, based on the measure of the ferrous ion-ferrozine complex formation, was used as an indicator of tomato fruits antioxidant activity and the results were presented in Table 4, Table 5 and Table 6. Without detaching sub-optimal and optimal fertility samples, ferrous chelating activity data ranged from 10.2 to 16.4 mg EDTAE/g extract, confirming the highest and lowest average value respectively for LowN and AMF + LowN application (Table 4). In addition, a markedly increase for sub-optimal fertility values (15.2 ± 1.4 mg EDTAE/g extract) was revealed, when compared to optimal fertility (12.8 ± 2.6 mg EDTA/g extract).

### 3.8. Enzyme Inhibitory Activity

The results for the inhibitory activity of tomato samples on α-amylase, AChE, BChE and tyrosinase were depicted in Appendix A. In general, no statistical difference between enzyme inhibition treatments was observed when comparing sub-optimal and optimal fertility (Table 4). Particularly, anti-α-Amylase activity exhibited the same values for all the treatments (0.2 ± 0.1 mmol ACAE/g extract). However, AMF + *T. atroviride* + LowN and control samples values tended to be great both for BChE (2.7 ± 0.5 and 2.7 ± 0.7 mg GALAE/g extract) and tyrosinase (57.4 ± 4.9 and 56.9 ± 6.1 mg KAE/g extract), while AMF + T. atroviride + Low N and AMF treatments confirmed the highest AChE inhibitory activity (2.4 ± 0.1 and 2.4 ± 0.2 mg GALAE/g extract). Except for tyrosinase, farm factor was crucial to determine dissimilarity between sub-optimal and optimal fertility sample, indicating a greater average value for cholinesterase (AChE and BChE) and Tyrosinase, respectively. Nevertheless, Table 5 and Table 6 revealed a different trend within two farms: within sub-optimal fertility, AChE and tyrosinase were significantly affected by treatment, while no differences were observed for any enzyme inhibitory activity within optimal fertility. 

### 3.9. Metabolomics Untargeted Analysis of the Phenolic Composition of Tomato Fruits

The metabolomic analysis with the UHPLC/QTOF investigated the phenolic compositions of tomato fruit sample subjected to different biostimulant treatments. Overall, when considering all the samples analyzed, 271 phenolic compounds were putatively annotated (Appendix A), mostly consisting of flavonoids (115). Moreover, 21 lignans, 62 phenolic acids and nine stilbenes were identified, while alkylmethoxyphenol, alkylphenol, curcuminoid, furanocoumarin, hydroxybenzaldehyde, hydroxybenzochetone, hydroxybenzoketone, hydroxycinnamaldehyde, hydroxycoumarin, hydroxyphenylpropene, methoxyphenol, naphtoquinone, phenolic terpene and tyrosol compounds were grouped together as “Other polyphenols” class (64).

An unsupervised multivariate approach, consisting of a fold-change-based hierarchical clustering, was carried out in order to gain insight into the biochemical processes of tomato fruits which appeared to be regulated following treatments. The analysis showed a clear division between sub-optimal and optimal fertility tomatoes, describing the farm as the unique factor affecting samples arrangement, since no clusters were formed for treatment factor (Figure 1).

Consequently, the hierarchical clustering analysis was repeated keeping the two farm samples separated in order to better achieve similarities and distances across treatments within the same pedoclimatic conditions (Figure 2). Regarding sub-optimal fertility, control samples were clustered together with AMF-treated tomatoes but highlighted a markedly different metabolomic profile from those samples obtained with the coupled action of AMF and *T. atroviride* under low N input (Figure 2A). Generally, treatments under low N rates appeared to be more distant from high N-treatments, thus showing LowN closer to AMF + LowN and AMF + *T. atroviride* + Low N, and further away from AMF and control. On the contrary, the nitrogen level within the treatment did not represent a discerning factor affecting clustering for optimal fertility, since no-separation N level-depending was achieved between samples (Figure 2B). Here, two clusters –respectively formed by Control and AMF + *T. atroviride* + Low N, and by AMF and AMF + LowN were observed.

Finally, two-ways ANOVA and Duncan test results for phenolic compound equivalents per class was provided in Appendix A. Here, significant differences among the optimal and sub-optimal fertility conditions were observed for each phenolic class and, consequently, Table 7 and Table 8 were provided in the main text to better get insight into the treatment effect in each site.

ANOVA and Duncan test results for phenolic compounds as equivalents per class were provided in Table 7 and Table 8. Regarding sub-optimal fertility, the treatment factor was statistically determinant for flavonoid class content, which varied from 2.9 to 8.1 mg eq. g^−1^ DM and showed the highest value (6.7 ± 1.4 mg eq. g^−1^ DM) with AMF + LowN application. Equally, a considerable treatment effect was remarked for lignan and other polyphenol classes, whose amounts were both mostly improved by AMF treatment (respectively 11.7 ± 3.3 and 37.7 ± 15.9 mg eq. g^−1^ DM). In contrast, in optimal fertility samples, none of the phenolic classes were significantly affected by the type of treatment.

## 4. Discussion

Finding new sustainable technologies to improve the functional and nutraceutical values of food products while improving yield and pomological traits has become a major research challenge due to ambitious objectives of the EU “Farm to fork” strategy [28]. In this context, the present study indicated that the use of biostimulants in agriculture may lead to a general increase of fruit quality-related compounds in *S. lycopersicum* L. On the contrary, we did not find a specific pattern in terms of tomato yield response to selected biostimulants.

Notably, mycorrhizal treatment revealed remarkable accumulations of carotenoids and phenols in tomato fruits, confirming the previous findings in literature [12,29,30]. Indeed, AMF have been proved to support plant accumulation of those secondary metabolites which are involved in the response to abiotic stresses and pathogens. This results in a concrete help for the plant to counteract the negative effects of the stress and, on the other hand, in the enhancement of the functional quality of edible plant parts. Specially carotenoids, implicated in plants’ defense mechanisms as antioxidants and photo-protecting molecules, play a key role against human cancer development, thus concurring to the nutraceutical quality of plant-based foods [31]. Likewise, phenolic compounds, involved in plant responses to environmental stress including wounding, pathogen attack, mineral deficiencies, and temperature stress, have been linked to a reduced risk of cardiovascular mortality for humans thanks to their high antioxidant potential [32,33].

More in details, our study highlights that a synergic biostimulant effect may be observed at the field level with the coupled inoculation of AMF and *T. atroviride*, which revealed increases in concentration for most compounds at higher rate than those due to single mycorrhizal application in our experiment. It has been reported that *T. atroviride* acts as a biocontrol agent against many aerial and soilborne plant pathogens, by activating different mechanisms, including competition for nutrients, production of useful secondary metabolites, modification of the rhizosphere, and mycoparasitism [34]. Interestingly, the inoculation of *T. atroviride* strain P1 on tomato plants exhibited a negative impact on the development of the noctuid moth *Spodoptera littoralis* and on the aphid *Macrosiphum euphorbiae* longevity, such as suppressed the phytopathogen *Phytophthora cinnamomic* [35,36]. Our results corroborate recent research, which previously showed the synergetic potential of AMF and *T. atroviride* co-inoculation, with increased plant growth, yield, nutrient uptake and stress tolerance [8,37,38]. Trichoderma-plant associations take place following the fungus secretion of proteins which are recognized by plant receptors. Successively, the following transient suppression of plant defenses promotes the Trichoderma penetration and, in case of co-inoculation, the concomitant access to AMF [39,40]. Similarly, it has been ascribed to AMF a corresponding help in Trichoderma conidia germination [41].

In the present study, AMF + *T. atroviride* + Low N- treated samples of sub-optimal fertility showed higher values of carotenoid concentrations—including B-carotene, z-carotene, all-e-y carotene, 13-z-lycopene and all-trans-lycopene—and total phenolic content. Similarly, the same treatment highlighted the strongest antioxidant abilities in DPPH, ABTS, phosphomolybdenum, reducing power and enzyme inhibitory assays. This suggests that the synergic biostimulant effect of AMF and *T. atroviride* is highly related to soil fertility status (as revealed here by our initial analyses on soil OM and total N), as confirmed by our results on optimal fertility showing no treatment differences in fertile soils.

Plant N uptake is greatly aided by mutualistic association with AMF, which grow and extend their hyphae in the surrounding soil, in exchange for photosynthetic carbon (C) from their plant hosts. Particularly, in addition to acquiring N in the two predominant forms, nitrate (NO_3_^−^) and ammonium (NH4^+^ ), AMF absorb and translocate N beyond the depletion zones of plant rhizosphere [42]. The small diameters of the mycorrhizal hyphae allow the fungus to efficiently penetrate soil micropores and uptake the inorganic forms of N which are released through microbial decomposition processes [43]. As a consequence, many studies have indicated an increased inflow of N (and other nutrients) in mycorrhizal root and, consequently, associated plants have been shown higher N concentrations than non-mycorrhized plants [44,45,46].

Nevertheless, N availability in soil can influence the relationship between plant and the fungus which reveals, as its concentration increases, a remarkable decrease in plant dependence on mycorrhizal symbiosis [47]. The effects of N-fertilization on AMF have mostly been assessed through changes in colonization intensity, diversity, and physiological functions of mycorrhizal fungi [48,49,50]. Interestingly, the meta-analysis of [51] reported an overall decrease of 15% in colonization percentage per length of root under N supply. Likely, a substantial decrease in soil AMF alpha diversity following N application has been thoroughly illustrated [52,53].

Moreover, AMF inoculation has been proved to help plant growth under low N conditions, modulating plant response at physiological, molecular, and metabolic level, resulting in secondary metabolites accumulation in the host plants [54]. Contrary, limited benefits of mycorrhizal inoculation for agricultural production have been found under high levels of N-fertilization, denoting a better promising in low-quality lands [47,55].

This evidence was strengthened by our study, where results from mycorrhized fruits showed evidence of improved levels of carotenoid and bioactive compound exclusively in sub-optimal fertility. In the light of this, we confirmed the plant stronger benefit from a symbiotic relationship with AMF in soil nutrient-scarce environments than in soil nutrient-rich environments [56,57,58,59].

## 5. Conclusions

The present field experiment suggested the single and coupled use of microbial biostimulants to improve the quality of tomato fruits. However, the results obtained highlighted a strong dependence of the biostimulant effect on soil fertilization, reflecting significant increment in antioxidant properties only under sub-optimal conditions. Neverthess, future research should be conducted with multi-year trials to strengthen these results, by assessing the biostimulant effect under different pedo-climatic conditions. Moreover, further and more in-depth studies should be carried out to fully understand the molecular and biochemical processes underlying the plant-fungi associations and the resulting changes in the functional value of fruits.

## Figures and Tables

**Figure 1 antioxidants-12-00520-f001:**
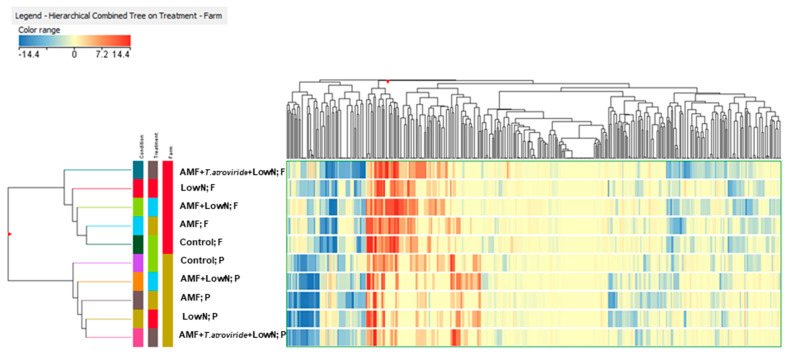
Unsupervised hierarchical cluster analysis (Euclidean distance; linkage rule: Ward) of tomato fruits phenolic profiles amid Control, LowN, AMF, AMF + LowN and AMF + *T. atroviride* + LowN treatments at optimal (P) and sub-optimal (F) fertility. Metabolites were obtained by UHPLC −ESI/QTOF−MS untargeted analysis, and their intensities were used to create the fold−change heatmap provided here.

**Figure 2 antioxidants-12-00520-f002:**
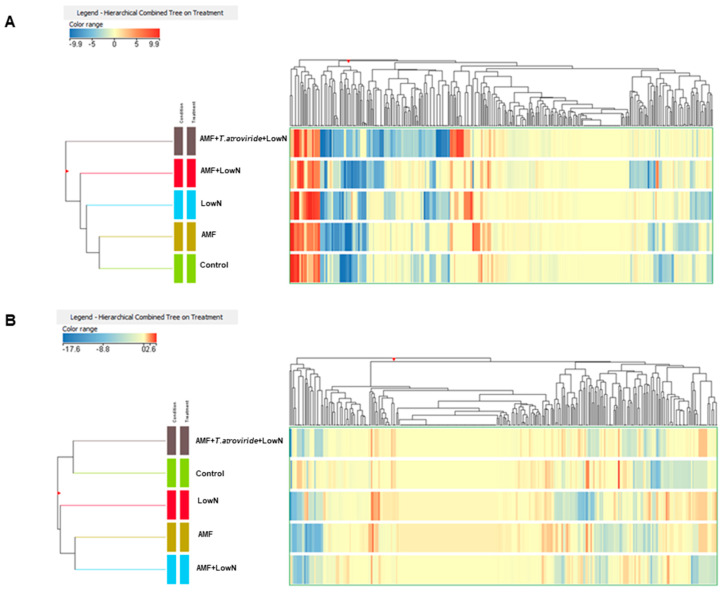
Unsupervised hierarchical cluster analysis (Euclidean distance; linkage rule: Ward) of sub-optimal fertility (**A**) and optimal fertility (**B**) tomato fruits phenolic profiles under LowN, AMF, AMF + LowN, AMF + T. atroviride + Low N and Control conditions. Metabolites were obtained by UHPLC−ESI/QTOF−MS untargeted analysis, and their intensities were used to create the fold −change heatmap provided here.

**Table 1 antioxidants-12-00520-t001:** Yield parameters for tomato grown under low nitrogen fertilization regimen, treated with different biostimulants at the site with sub-optimal soil fertility. The data were elaborated by one-way analysis and Duncan test.

	Yield Parameters				
	Marketable (t/ha)	Immature (t/ha)	Rotten (t/ha)	Apical Rot (t/ha)	Total Yield (t/ha)
Treatment (T)					
LowN	68.0	5.4	1.4	1.2 ^a^	76.1
AMF	65.8	3.8	1.1	0.8 ^a^	71.5
AMF + LowN	70.7	4.8	1.4	1.5 ^a^	78.4
AMF + *T. atr.* + LowN	71.9	10.1	1.4	1.5 ^a^	84.9
Control	70.2	7.9	0.8	2.1 ^b^	81.0
Significance					
T (df = 5) Residuals (df = 18)	n.s.	n.s.	n.s.		n.s.

Different letters show significant difference at the 0.05 probability level. n.s.: non-significant.

**Table 2 antioxidants-12-00520-t002:** Yield parameters for tomato grown under low nitrogen fertilization regimen, treated with different biostimulants at the site with optimal soil fertility. The data were elaborated by one-way analysis and Duncan test.

	Yield Parameters				
	Marketable (t/ha)	Immature (t/ha)	Rotten (t/ha)	Apical Rot (t/ha)	Total Yield (t/ha)
Treatment (T)					
LowN	82.7 ^b^	13	2.1	0.4	98.2 ^b^
AMF	88.8 ^a^	12.3	3.1	0.5	104.7 ^a^
AMF + LowN	78.6 ^b^	7.4	1.9	0.5	88.5 ^b^
AMF + *T. at*r. + LowN	88.0 ^a^	10.2	2.7	1.0	101.9 ^a^
Control	89.0 ^a^	12.1	3.1	0.8	104.9 ^a^
Significance					
T (df = 5) Residuals (df = 18)		n.s.	n.s.	n.s.	

Different letters show significant difference at the 0.05 probability level. n.s.: non-significant.

**Table 3 antioxidants-12-00520-t003:** Carotenoid content in tomato fruits from plants grown under low nitrogen fertilization regimen, treated with different biostimulants at the site with sub-optimal soil fertility. The data were elaborated by one-way analysis and Duncan test.

Source of Variance	Phytoene mg/100 g	Phytofluene mg/100 g	Z-b-carotene mg/100 g	B-carotene mg/100 g	Z-carotene mg/100 g	All-E-y-carotene mg/100 g	13-z-lycopene mg/100 g	7-z-lycopene mg/100 g	9-z-lycopene mg/100 g	Cis-Lycopene mg/100 g	All-Trans-Lycopene mg/100 g
Thesis (T)											
LowN	3.278 ± 0.649 ^a^	1.233 ± 0.015 ^a^	0.131 ± 0.141	0.598 ± 0.262	0.020 ± 0.001	0.063 ± 0.001 ^a^	0.126 ± 0.061	0.151 ± 0.06	0.020 ± 0.001 ^a^	0.247 ± 0.023	11.982 ± 4.985
AMF	1.821 ± 1.136 ^c^	0.599 ± 0.138 ^c^	0.044 ± 0.021	0.433 ± 0.103	0.006 ± 0.009	0.045 ± 0.004 ^b^	0.103 ± 0.020	0.112 ± 0.022	0 ^b^	0.175 ± 0.043	9.451 ± 1.763
AMF + LowN	2.521 ± 0.475 ^b^	0.839 ± 0.141 ^b^	0.033 ± 0.010	0.543 ± 0.088	0.008 ± 0.014	0.063 ± 0.010 ^a^	0.142 ± 0.037	0.175 ± 0.036	0.008 ± 0.012 ^b^	0.256 ± 0.044	12.415 ± 1.648
AMF + *T. atr*. + LowN	2.599 ± 0.424 ^b^	0.856 ± 0.293 ^b^	0.038 ± 0.011	0.647 ± 0.243	0.021 ± 0.021	0.063 ± 0.022 ^a^	0.145 ± 0.052	0.154 ± 0.037	0.007 ± 0.012 ^b^	0.233 ± 0.051	12.586 ± 1.511
Control	3.426 ± 0.424 ^a^	1.042 ± 0.099 ^ab^	0.082 ± 0.079	0.486 ± 0.189	0.014 ± 0.021	0.071 ± 0.015 ^a^	0.135 ± 0.053	0.191 ± 0.074	0.008 ± 0.012 ^b^	0.215 ± 0.017	10.435 ± 4.551
Significance											
T (df = 4)—*p* value	3.23 × 10-5	9.63 × 10-6	0.264	0.337	0.374	0.0297	0.556	0.106	0.0178	0.184	0.184
Residuals (df = 25) F value	10.78	12.58	1.397	1.197	1.109	3.202	0.768	2.137	3.651	1.687	1.687

Different letters show significant difference at the 0.05 probability level.

**Table 4 antioxidants-12-00520-t004:** Carotenoid content in tomato fruits from plants grown under low nitrogen fertilization regimen, treated with different biostimulants at the site with optimal soil fertility. The data were elaborated by one-way analysis and Duncan test.

Source of Variance	Phytoene mg/100 g	Phytofluene mg/100 g	Z-b-carotene mg/100 g	B-carotene mg/100 g	Z-carotene mg/100 g	All-E-y-carotene mg/100 g	13-z-lycopene mg/100 g	7-z-lycopene mg/100 g	9-z-lycopene mg/100 g	Cis-Lycopene mg/100 g	All-Trans-Lycopene mg/100 g
Thesis (T)											
LowN	3.718 ± 0.649	1.042 ± 0.099 ^a^	0.315 ± 0.022	0.436 ± 0.070 ^bc^	0 ^b^	0.079 ± 0.008	0.215 ± 0.020	0.262 ± 0.037	0.024 ± 0.002	0.418 ± 0.0766	16.781 ± 1.797
AMF	3.310 ± 1.182	1.056 ± 0.280 ^ab^	0.255 ± 0.055	0.332 ± 0.099 ^c^	0.008 ± 0.013 ^ab^	0.061 ± 0.022	0.170 ± 0.069	0.182 ± 0.073	0.015 ± 0.012	0.274 ± 0.122	12.201 ± 3.562
AMF + LowN	2.611 ± 0.291	0.827 ± 0.088 ^a^	0.279 ± 0.034	0.573 ± 0.082 ^a^	0.020 ± 0.002 ^a^	0.072 ± 0.014	0.179 ± 0.039	0.191 ± 0.043	0.019 ± 0.003	0.315 ± 0.085	13.694 ± 2.439
AMF + *T. atr.* + LowN	2.763 ± 1.997	1.051 ± 0.087 ^ab^	0.295 ± 0.018	0.647 ± 0.243 ^c^	0.009 ± 0.014 ^ab^	0.072 ± 0.011	0.200 ± 0.206	0.206 ± 0.005	0.021 ± 0.001	0.349 ± 0.027	14.549 ± 0.225
Control	3.563 ± 1.263	1.190 ± 0.263 ^a^	0.310 ± 0.041	0.552 ± 0.158 ^ab^	0.009 ± 0.014 ^b^	0.079 ± 0.021	0.220 ± 0.060	0.278 ± 0.113	0.023 ± 0.006	0.468 ± 0.235	16.072 ± 4.187
Significance											
T (df = 4)—*p* value	0.0668	0.0267	0.514	0.00209	0.0265	0.65	0.282	0.0599	0.121	0.102	0.0632
Residuals (df = 25) F value	2.517	3.294	2.735	5.709	3.302	0.624	1.344	2.607	2.027	2.171	2.563

Different letters show significant difference at the 0.05 probability level.

**Table 5 antioxidants-12-00520-t005:** Antioxidant assays and enzyme inhibition properties in tomato fruits from plants grown under low nitrogen fertilization regimen, treated with different biostimulants at the site with sub-optimal soil fertility. The data were elaborated by one-way analysis and Duncan test.

Source of Variance	Total Phenolic Content mgGAE/g	Total Flavonoid Content mgRE/g	DPPH mgTE/g	ABTS mgTE/g	CUPRAC mgTE/g	FRAP mgTE/g	Metal Chelating mgEDTAE/g	Phosphomolybdenum mmolTE/g	AChe inh. mgGALAE/g	BChe inh. mgGALAE/g	Tyrosinase mgKAE/g	Amylase mmolACAE/g
Thesis (T)												
LowN	11.9 ± 2.0	0.8 ± 0.1 ^b^	1.6 ± 0.4	23.0 ± 2.9	33.3 ± 3.8 ^ab^	20.4 ± 1.1 ^ab^	15.4 ± 1.4	0.8 ± 0.1 ^ab^	2.3 ± 0.1	2.4 ± 0.3	56.9 ± 4.9 ^c^	0.21 ± 0.0
AMF	11.9 ± 2.2	0.9 ± 0.1 ^ab^	1.4 v 1.3	21.6 ± 4.4	33.4 ± 5.0 ^ab^	20.6 ± 2.5 ^ab^	13.6 ± 1.8	0.9 ± 0.1 ^a^	2.4 ± 0.1	2.2 ± 0.6	54.9 ± 4.2 ^bc^	0.20 ± 0.0
AMF + LowN	10.6 ± 1.1	0.9 ± 0.2 ^a^	1.6 ± 0.7	20.3 ± 2.5	29.5 ± 1.9 ^b^	19.6 ± 0.8 ^b^	14.0 ± 2.8	0.7 ± 0.1 ^b^	2.2 ± 0.1	2.4 ± 0.4	56.9 ± 6.2 ^abc^	0.21 ± 0.0
AMF + *T. atr*. + LowN	12.9 ± 2.9	0.8 ± 0.1^ab^	3.4 ± 3.7	24.5 ± 8.3	37.4 ± 7.6^a^	23.6 ± 6.3a	15.2 ± 1.6	0.9 ± 0.2^a^	2.4 ± 0.1	2.3 ± 0.4	62.2 ± 3.1^ab^	0.21 ± 0.0
Control	11.2 ± 1.3	0.8 ± 0.1^b^	1.2 ± 0.4	21.2 ± 0.8	31.2 ± 1.8^b^	19.2 ± 1.0^b^	14.4 ± 2.5	0.8 ± 0.1^ab^	2.2 ± 0.1	2.7 ± 0.9	63.5 ± 4.6^a^	0.21 ± 0.0
Significance												
T (df = 4)—*p* value	0.166	0.0452	0.0964	0.331	0.0114	0.0431	0.282	0.0483	0.00116	0.551	0.0315	0.104
Residuals (df = 40) F value	1.712	2.681	2.118	1.188	3.73	2.717	1.313	2.632	5.599	0.77	2.951	2.064

Different letters show significant difference at the 0.05 probability level.

**Table 6 antioxidants-12-00520-t006:** Antioxidant assays and enzyme inhibition properties in tomato fruits from plants grown under low nitrogen fertilization regimen, treated with different biostimulants at the site with optimal soil fertility. The data were elaborated by one-way analysis and Duncan test.

Source of Variance	Total Phenolic Content mgGAE/g	Total Flavonoid Content mgRE/g	DPPH mgTE/g	ABTS mgTE/g	CUPRAC mgTE/g	FRAP mgTE/g	Metal Chelating mgEDTAE/g	Phosphomolybdenum mmolTE/g	AChe inh. mgGALAE/g	BChe inh. mgGALAE/g	Tyrosinase mgKAE/g	Amylase mmolACAE/g
Thesis (T)												
LowN	14.4 ± 1.7 ^a^	0.9 ± 0.1 ^bc^	3.2 ± 1.6 ^a^	29.8 ± 1.7 ^a^	35.6 ± 5.7	21.8 ± 2.2 ^a^	14.9 ± 1.4 ^a^	0.9 ± 0.1	2.4 ± 0.2	2.8 ± 0.7	53.9 ± 4.7	0.19 ± 0.0
AMF	12.7 ± 0.4 ^b^	0.8 ± 0.1 ^c^	1.4 ± 05 ^b^	23.2 ± 1.4 ^c^	32.5 ± 1.6	19.5 ± 0.6 ^bc^	12.5 ± 1.2 ^b^	0.9 ± 0.1	2.4 ± 0.2	3.0 ± 0.7	54.7 ± 3.5	0.18 ± 0.0
AMF + LowN	12.4 ± 0.8 ^b^	1.0 ± 0.2 ^ab^	1.5 ± 0.2 ^b^	23.6 ± 0.9 ^c^	31.6 ± 2.5	19.2 ± 0.9 ^c^	11.5 ± 1.7 ^b^	0.8 ± 0.1	2.3 ± 0.1	2.9 ± 0.4	53.8 ± 3.2	0.19 ± 0.0
AMF + T.atr. + LowN	12.5 ± 1.7 ^b^	0.8 ± 0.2 ^bc^	1.6 ± 1.4 ^b^	24.8 ± 3.1 ^bc^	32.9 ± 3.6	19.7 ± 1.4 ^bc^	12.7 ± 2.7 ^b^	0.9 ± 0.1	2.4 ± 0.1	3.1 ± 0.3	53.9 ± 4.1	0.19 ± 0.0
Control	13.4 ± 0.6 ^ab^	1.2 ± 0.3 ^a^	1.8 ± 0.1 ^b^	27.1 ± 3.8 b	34.7 ± 2.1	20.7 ± 1.2 ab	12.4 ± 1.7 b	0.9 ± 0.1	2.4 ± 0.1	2.7 ± 0.6	52.9 ± 2.3	0.19 ± 0.0
Significance												
T (df = 4)—*p* value	0.00368	0.00118	0.00286	3.69 x 10-6	0.117	0.00156	0.00586	0.96	0.581	0.686	0.29	0.833
Residuals (df = 40) F value	4.619	5.561	4.824	11.12	1.976	5.323	4.247	0.153	0.723	0.571	1.29	0.364

Different letters show significant difference at the 0.05 level.

**Table 7 antioxidants-12-00520-t007:** Semi-quantitative content of different phenolic classes in tomato fruits from plants grown under low nitrogen fertilization regimen, treated with different biostimulants at the site with sub-optimal soil fertility. The data are presented as cumulate abundance of individual compounds gained from untargeted metabolomics profiling and were elaborated by one-way analysis and Duncan test.

Source of Variance	Flavonoids mg eq. g^−1^ DM	Lignans mg eq. g^−1^ DM	Other Polyphenols mg eq. g^−1^ DM	Phenolic Acids mg eq. g^−1^ DM	Stilbenes mg eq. g^−1^ DM
Thesis (T)					
LowN	4.7 ± 1.8 ^b^	13.2 ± 1.2 ^a^	52.6 ± 14.9 ^a^	27.7 ± 4.5	0.4 ± 0.2
AMF	5.2 ± 0.8 ^ab^	11.7 ± 3.3 ^ab^	37.7 ± 15.9 ^ab^	28.2 ± 5.6	0.4 ± 0.1
AMF + LowN	6.7 ± 1.4 ^a^	6.4 ± 1.8 ^c^	22.1 ± 4.6 ^b^	21.9 ± 8.7	0.5 ± 0.1
AMF + *T. atr*. + LowN	4.5 ± 1.1 ^b^	8.4 ± 3.7 ^bc^	24.8 ± 13.3 ^b^	17.1 ± 9.7	0.5 ± 0.1
Control	6.1 ± 0.9 ^ab^	11.4 ± 2.7 ^ab^	30.9 ± 14.1 ^b^	21.9 ± 8.7	0.5 ± 0.1
Significance					
T (df = 4)—*p* value	0.024	0.00189	0.00382	0.106	0.566
Residuals (df = 25) F value	3.388	5.815	5.098	2.136	0.752

Different letters show significant difference at the 0.05 level.

**Table 8 antioxidants-12-00520-t008:** Semi-quantitative content of different phenolic classes in tomato fruits from plants grown under low nitrogen fertilization regimen, treated with different biostimulants at the site with optimal soil fertility. The data are presented as cumulate abundance of individual compounds gained from untargeted metabolomics profiling and were elaborated by one-way analysis and Duncan test.

Source of Variance	Flavonoids mg eq. g^−1^ DM	Lignans mg eq. g^−1^ DM	Other Polyphenols mg eq. g^−1^ DM	Phenolic Acids mg eq. g^−1^ DM	Stilbenesmg eq. g^−1^ DM
Thesis (T)					
LowN	3.5 ± 0.7	13.0 ± 1.9	55.0 ± 11.5	30.8 ± 7.2	0.5 ± 0.2
AMF	2.4 ± 0.3	12.8 ± 3.1	44.2 ± 16.2	32.4 ± 6.2	0.5 ± 0.1
AMF + LowN	3.1 ± 1.0	10.9 ± 4.9	31.9 ± 13.6	24.0 ± 11.7	0.5 ± 0.2
AMF + *T. atr*. + LowN	3.7 ± 1.0	11.2 ± 4.0	37.9 ± 19.3	27.3 ± 12.8	0.6 ± 0.1
Control	3.8 ± 1.4	12.7 ± 2.8	37.4 ± 15.7	27.7 ± 9.6	0.6 ± 0.2
Significance					
T (df = 4)—*p* value	0.105	0.36	0.291	0.179	0.709
Residuals (df = 25) F value	2.148	1.142	1.316	1.714	0.538

Different letters show significant difference at the 0.05 probability level.

## Data Availability

Raw data are enclosed as Appendix A.

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
