# Peer review of "The Functional Profile and Antioxidant Capacity of Tomato Fruits Are Modulated by the Interaction between Microbial Biostimulants, Soil Properties, and Soil Nitrogen Status"

_antioxidants, 2023, doi:10.3390/antiox12020520_

Round 1

Reviewer 1 Report

Authors studied the effect of single or coupled action of biostimulants, associated with either optimal or reduced nitrogen application, on the functional quality of tomato berries.

The authors conducted experiments in two field conditions in 2020, but due to data variability, the reliability of the agricultural and metabolic data was not well understood. What are the flavonoids, lignans and small-molecular-weight phenolics in the abstract? Please show the specific metabolites. Authors described on abstract like “However, a strong dependence of the biostimulant effect on nitrogen availability was also noticed, reflecting significant increment in antioxidant activity under sub-optimal fertility conditions and low nitrogen levels.” Which data is that? I would like to request that you describe a summary showing specific data on whether flavonoids, lignans and small-molecular-weight phenolics, etc. increased or decreased in comparison to the control group. If possible, I recommend the multi-year trials to verify the impact of biostimulants. As a minor point, is Fig. 2 an divided view of Fig. 1?

Author Response

dear Reviewer,

thanks for the critical revision. We are including a point-to-point rebuttal for each criticism that has been raised up from the review process.

regards

Reviewer 2 Report

Comments

In the Introduction section, the author did not explain why R. irregulare and F. mosseae were used as biostimulators. Have those biostimulators been tested in experiments conducted earlier by other scientists?

·      Line 112

How were carotenoids extracted? What chromatographic system was used for carotenoid separation? During the UPLC/HPLC analysis, what gradient programme was used?

The parameters of the column are not specified.

·      Line 124

All results were calculated per g of extract; however, the author did not describe the method for extract preparation, the mass of the plant material and the volume of solutions. Was purification or fractionation performed?

·      Line 145

What were the parameters of the mass spectrometer set? What energy?

·      Line 147

What chromatographic column was used for polyphenol separation?

What was the gradient programme used for the mobile phases?

Were formic acid and/or ammonium formate added to mobile phases? If so, what concentration?

·      Tables 1 and 2

Why did the author title Tables 1 and 2 as statistical analysis, while those tables show results of tomato fruit yield?

Moreover, why are the yield results for sub-optimal and optimal fertilisation present separately? If the author compares these data further, then the author should present the results in one table and calculate statistics between all results (e.g., all results for immature yield after sub-optimal and optimal fertilization).

·      A similar question should be asked concerning the next tables, 3 and 4, 5 and 6, 7 and 8.

·      What does the T parameter (in all tables) mean? If the author analyzed the ANOVA test, then the estimation returns F, df, and p statistics. Please specify these statistics in the tables.

·      The author presented many results in tables and as supplementary materials; however, some results were not discussed.

Please eliminate unnecessary results (e.g., total refractometric - column 1 and optical residue - column 11); on the other hand, thoroughly discuss all results presented in the manuscript.

Do not duplicate the same results in tables (e.g., immature yield - expressed as t/ha and %, columns 3 and 8 in table 1).

·      The discussion is poor and there is no novelty. It should be rewritten again.

Author Response

(The authors gave the same response as above.)

Round 2

Reviewer 2 Report

There is a doubt; please answer why the positive mode of ESI was used during polyphenols UHPLC-ESI/QTOF-MS analysis since negative mode is generally applied.

If you used positive mode, please specify what molecular ion you have identified, e.g., for caffeic acid. In Table S3, you show a mass of 183 for caffeic acid; what does that mass mean, or what molecular ion? 

Moreover,

Please change titles for all tables because these tables contain results for  yield and/or content of chemicals within tomato fruits. The statistical analysis is not the primary focus of the table, but it only illustrates the differences in the yield.

Therefore, table 3 should be titled "Carotenoid content in tomato fruits during sub-optimal fertility" instead of title: " One-way analysis and Duncan test for carotenoid content in tomato fruits of sub-optimal fertility".

Author Response

dear Reviewer,

thanks for the critical revision. A point-to-point response to each comment is provided in the attached file.
